# Association between chiropractic spinal manipulation and cauda equina syndrome in adults with low back pain: Retrospective cohort study of US academic health centers

Robert J. Trager[1,2,3]*, Anthony N. Baumann[4,5], Jaime A. Perez[6], Jeffery A. Dusek[2], Romeo-Paolo T. Perfecto[3,7,8], Christine M. Goertz[7,8,9]

1 Connor Whole Health, University Hospitals Cleveland Medical Center, Cleveland, Ohio, United States of America, 2 Department of Family Medicine and Community Health, Case Western Reserve University School of Medicine, Cleveland, Ohio, United States of America, 3 Department of Biostatistics and Bioinformatics Clinical Research Training Program, Duke University School of Medicine, Durham, North Carolina, United States of America, 4 Department of Rehabilitation, University Hospitals Cleveland Medical Center, Cleveland, Ohio, United States of America, 5 College of Medicine, Northeast Ohio Medical University, Rootstown, Ohio, United States of America, 6 Clinical Research Center, University Hospitals Cleveland Medical Center, Cleveland, Ohio, United States of America, 7 Department of Orthopaedic Surgery, Duke University, Durham, North Carolina, United States of America, 8 Duke Clinical Research Institute, Durham, North Carolina, United States of America, 9 Robert J. Margolis, MD, Center for Health Policy, Duke University, Durham, North Carolina, and Washington, District of Columbia, United States of America

* Robert.Trager@duke.edu

**Data Availability Statement:** The minimal, de-identified, aggregated data used to describe baseline characteristics, our primary outcome, and

## Abstract

### Background

Cauda equina syndrome (CES) is a lumbosacral surgical emergency that has been associated with chiropractic spinal manipulation (CSM) in case reports. However, identifying if there is a potential causal effect is complicated by the heightened incidence of CES among those with low back pain (LBP). The study hypothesis was that there would be no increase in the risk of CES in adults with LBP following CSM compared to a propensity-matched cohort following physical therapy (PT) evaluation without spinal manipulation over a three-month follow-up period.

### Methods

A query of a United States network (TriNetX, Inc.) was conducted, searching health records of more than 107 million patients attending academic health centers, yielding data ranging from 20 years prior to the search date (July 30, 2023). Patients aged 18 or older with LBP were included, excluding those with pre-existing CES, incontinence, or serious pathology that may cause CES. Patients were divided into two cohorts: (1) LBP patients receiving CSM or (2) LBP patients receiving PT evaluation without spinal manipulation. Propensity score matching controlled for confounding variables associated with CES.

plot propensity score density curves and cumulative incidence are available in figshare (https://doi.org/10.6084/m9.figshare.24654396).

**Funding:** This project was supported by the Clinical and Translational Science Collaborative of Northern Ohio which is funded by the National Institutes of Health, National Center for Advancing Translational Sciences, Clinical and Translational Science Award grant, UM1TR004528. The content is solely the responsibility of the authors and does not necessarily represent the official views of the National Institutes of Health.

**Competing interests:** Robert J. Trager acknowledges that he has received royalties as the author of two texts on the topic of sciatica. The authors have declared no competing interests.

## Results

67,220 patients per cohort (mean age 51 years) remained after propensity matching. CES incidence was 0.07% (95% confidence intervals [CI]: 0.05–0.09%) in the CSM cohort compared to 0.11% (95% CI: 0.09–0.14%) in the PT evaluation cohort, yielding a risk ratio and 95% CI of 0.60 (0.42–0.86; $p$ = .0052). Both cohorts showed a higher rate of CES during the first two weeks of follow-up.

## Conclusions

These findings suggest that CSM is not a risk factor for CES. Considering prior epidemiologic evidence, patients with LBP may have an elevated risk of CES independent of treatment. These findings warrant further corroboration. In the meantime, clinicians should be vigilant to identify LBP patients with CES and promptly refer them for surgical evaluation.

## Introduction

The cauda equina is a bundle of nerve roots arising from the spinal cord at the upper lumbar spine [1, 2]. Compression of these nerve roots, typically by a disc herniation [1, 3], can cause cauda equina syndrome (CES). Signs and symptoms of CES include one or more of the following (1) bladder/bowel dysfunction, (2) reduced saddle area sensation or (3) sexual dysfunction [4], and potentially low back pain (LBP) or lower extremity symptoms [4]. CES with neurological deficits is a medical emergency and surgical intervention is recommended within 48 hours to prevent permanent damage [5]. While CES is rare among asymptomatic individuals (0.6 cases per 100,000 per year), it is more common among those with LBP, affecting 270 per 100,000 (0.27%) per year in secondary care settings [6].

CES has given rise to a substantial number of medicolegal cases within both the chiropractic and physical therapy (PT) professions, perhaps because these clinicians commonly manage LBP [7–9]. It is thought that some of these cases occur because clinicians fail to recognize evolving CES features and refer appropriately, leading to a delay in diagnosis and surgery [5, 7–10]. However, in some instances, the degree to which the clinician was negligent is unclear as early identification of CES is compounded by potentially mild or gradually-developing symptoms [10]. For example, a broad review of medicolegal CES cases found that only 27% of patients initially presented with loss of bowel or bladder function [5].

In addition to the possibility of missed CES cases, concerns have been raised regarding documented cases of CES that occurred following chiropractic spinal manipulation (CSM). It has been hypothesized that CSM increases spinal loading, which potentially worsens the type of pre-existing disc injuries that can lead to CES [11, 12]. However, others have suggested that CSM is likely not a meaningful risk factor for CES due to its rarity following CSM when compared to the millions of CSM treatments administered annually [13–15]. In fact, a retrospective study including 54,846 patients of all ages and with various chief complaints found no instances of CES following 960,140 sessions of CSM [16]. However, to the authors' knowledge, no additional large studies have examined this issue.

Chiropractors are increasingly sought by patients in the US for the treatment of LBP [17]. A recent study based on insurance claims revealed that chiropractors were among the most commonly visited healthcare providers for new episodes of LBP, ranking second only to primary care physicians (25.2% of episodes with primary care versus 24.8% with a chiropractor) [18].

Furthermore, chiropractors use spinal manipulation more frequently than any other type of clinician [18].

Half of chiropractic patients have LBP, [19] with a subset of these patients having lumbar disc herniation [20]. Although CES is a rare event, lumbar disc herniation is its most common cause [2] and is also frequently present among those with LBP [21]. Accordingly, chiropractors may encounter patients who have a heightened risk of developing CES, as these clinicians treat those with LBP and disc disorders [11, 14].

Considering CSM is commonly used for LBP, wherein underlying disc degeneration may pose a risk factor for CES [11, 14], researchers have emphasized the importance of studying the potential association between CSM and CES [14, 22]. Mild adverse events related to CSM, such as transient soreness, are accepted to be common and occur in 23–83% of patients [23]. However, less is known regarding the potential for CSM to cause CES, as the existing literature on the topic is mostly derived from individual case reports [11, 12, 14].

The frequency with which chiropractors encounter undiagnosed CES is unclear. In a retrospective study of 7,221 patients presenting to chiropractors for new-onset LBP, no patients met the criteria for CES [24]. Only a handful of case reports have described chiropractors identifying CES [25–28]. However, one study estimated that 0.1% of 1.6 million patients presenting for PT were recognized as having CES [29]. Given the similarity of chiropractic and PT as conservative, nonpharmacologic secondary care entry points for LBP [18, 30], patients with CES could potentially present to either clinician type.

Given the possibility of harm raised by previous case reports, it is necessary to examine the potential association between receiving CSM and the risk of subsequent CES in adults with LBP. The achieved aim of this project was to test our hypothesis that adults with LBP receiving CSM have no significant increased risk of CES compared to those undergoing PT evaluation without spinal manipulation.

## Materials and methods

### Study design

This study used a retrospective cohort design with active comparator features to reduce bias [31] and followed a registered protocol [32]. A visual representation of the study design is available in the supplementary material (S1 Fig). Study reporting follows the Strengthening the Reporting of Observational Studies in Epidemiology (STROBE) guideline [33]. We included patients starting 20 years' prior to the query date (July 30, 2023) to maximize sample size. The inclusion window ended three months prior to the query date to allow identification of CES during follow-up.

This study used fully anonymized, de-identified data and therefore was deemed Not Human Subjects Research by the University Hospitals Institutional Review Board (Cleveland, Ohio, USA, STUDY20230269), which also waived the need for consent. TriNetX is compliant with the Health Insurance Portability and Accountability Act (HIPAA) [34]. TriNetX only contains de-identified data as per the de-identification standard defined in Section §164.514 (a) of the HIPAA Privacy Rule. The TriNetX network contains data provided by participating healthcare organizations, each of which represents and warrants that it has all necessary rights, consents, approvals, and authority to provide the data to TriNetX under a Business Associate Agreement, so long as their name remains anonymous as a data source and their data are utilized for research purposes. The data shared through the TriNetX Platform are attenuated to ensure that they do not include sufficient information to facilitate the determination of which health care organization contributed which specific information about a patient.

## Setting and data source

Data were obtained from a US research network (TriNetX Inc., Cambridge, Massachusetts, USA) which includes health records data from over 105 million patients. The dataset includes 76 contributing health care organizations which are large, academic medical centers and their affiliated community hospitals and ambulatory offices. The data are routinely collected, related to patient care, include both insured and uninsured patients, and can be searched using standardized nomenclatures such as the International Classification of Diseases, 10th Edition codes (ICD-10) [34]. A central TriNetX team regularly examines the dataset for conformance, plausibility, and completeness [34]. The TriNetX software interconverts ICD-10 to ICD-9 codes in queries of older medical records [34].

Precise data regarding the characteristics of chiropractors and PTs in the included healthcare organizations (e.g., years of experience, additional training) was not available due to de-identification of the dataset. In general, US chiropractors must complete a doctoral-level degree and pass the National Chiropractic Board of Chiropractic Examiners examinations [35]. In addition, the chiropractic scope of practice is legally regulated [36], and each US state requires continuing education credits [37]. However, evidence suggests that only a minority of chiropractic and PT clinicians in the US are employed in a hospital-based practice setting such as those included in the TriNetX dataset [38, 39]. One study reported that chiropractors in hospital-based settings were a mean 21 years' post-graduation (minimum: 2 to maximum: 40) [40].

Natural language processing was used to bolster the identification of data items from clinical notes, using machine learning technology available within TriNetX (Averbis, Freiburg im Breisgau, DE). This feature includes mechanisms to understand negation (e.g., absence of a condition), intent, and context, thus aiding us in (1) excluding patients with prior CES as noted in free text in medical records, and (2) identifying new diagnoses of CES during follow-up. Prior studies have demonstrated that this software has acceptable accuracy, reliability, and agreement when compared to manual chart review for extracting clinical concepts related to diagnoses, laboratory values, medications, and symptoms [41, 42].

## Participants

**Eligibility criteria.** Patients were included having either localized or radiating LBP (i.e., radicular pain, sciatica), lumbar disc degeneration, or lumbar disc herniation via any of a combination of commonly used ICD-10 diagnosis codes [43, 44] (S1 Table). Patients were required to be at least 18 years of age, considering that CES is uncommon in the pediatric population [6] and we are unaware of any cases of CES following CSM in pediatric patients [11, 45]. Patients were included on the date of initial CSM or PT evaluation for LBP. Patients receiving CSM were further required to have the presence of a segmental dysfunction code for the thoracic or lumbopelvic regions (i.e., M99.02, M99.03, M99.04, M99.05) indicating that CSM was applied to any of these regions. While CES typically arises from the lumbosacral region, medicolegal reports have documented CES occurring after thoracic CSM [7]. Accordingly, patients receiving only cervical CSM were not included. To help ensure that patients' medical information was complete, patients were required to have a previous healthcare visit between one week and two years preceding the index date (CSM or PT evaluation).

Patients were excluded who had a previous diagnosis of CES, injury to the cauda equina, neuromuscular dysfunction of the bladder, urinary or fecal incontinence, bladder catheterization, and serious pathology of the lumbar spine (i.e., fracture, malignancy, infection, and bleeding disorders) which may cause CES [1, 4], and congenital abnormalities of the cauda equina such as tethered cord syndrome using an exclusion assessment window of six months (S2 Table). Considering other spinal manipulative therapies resemble CSM, such as manual

therapies (e.g., spinal mobilization) and osteopathic manipulation, patients receiving these treatments were excluded from the PT evaluation cohort.

## Variables

**Cohorts.** Patients were divided into two cohorts: (1) CSM; patients identified at the first co-occurrence of CSM (via Current Procedural Terminology codes 98940, 98941, and/or 98942) with a LBP diagnosis and (2) PT evaluation; those receiving a new patient evaluation or re-evaluation for LBP and not receiving any spinal manipulation. Patients in the CSM cohort received CSM in the thoracic, lumbar and/or sacroiliac/pelvic regions. PT evaluations were identified using Current Procedural Terminology codes (i.e., 97161, 97162, 97163, 97164). Current Procedural Terminology and ICD-10 codes used for inclusion and exclusion criteria can be found in the supplementary files (S1 and S2 Tables).

**Confounding variables.** Propensity score matching was used to reduce bias from confounding variables [31], matching variables present within one year preceding index date of inclusion and having a known association with CES (S3 Table): body mass index [46, 47], constipation [48], demographics (age, sex, race, ethnicity [49]), any emergency department visit [6], epidural steroid injection [50], lumbar/lumbosacral disc herniation with radiculopathy or sciatica [1, 3], lumbar spine advanced imaging [48] (e.g., magnetic resonance imaging or computed tomography), lumbar stenosis and spondylolisthesis [1, 47, 51, 52], and previous spine surgery [53].

**Outcome.** Considering the potential for a delayed diagnosis [3, 54, 55], identification of occurrences of CES was over a 90-day follow-up window commencing from the index date of CSM or PT evaluation. This duration aligns with a prior study which noted that the median time to CES diagnosis was 11 days (SD = 24), with a maximum of 90 days [55]. The need for a 90-day follow-up was further supported by research suggesting that CES develops gradually in older adults [52], and a review of medicolegal cases of CES after CSM in which CES was only immediate in one case [7]. Assessment included any occurrence of either a diagnosis of CES (ICD-10: G83.4) or injury to the cauda equina (ICD-10: S34.3) [56, 57]. To account for variability in diagnostic coding versus free-text charting, ascertainment of these diagnoses was further refined by natural language processing of narrative text appearing in patients' clinical charts.

The likelihood of lumbar spine surgery was not examined considering this outcome would not be specific to CES [5]. Other potential CSM-related adverse events were not examined such as spinal fracture or hematoma considering these conditions would require different selection criteria and propensity matched variables.

## Statistical methods

Using the TriNetX platform, baseline characteristics were compared using built-in independent-samples t-tests and Pearson chi-squared tests, and propensity score matching used Python's scikit-learn package (version 3.7 Python Software Foundation, Delaware, USA). Logistic regression was used to calculate propensity scores for patients, and greedy nearest-neighbor matching was performed using a 1:1 ratio with a caliper of 0.1 pooled standard deviations [58]. The risk ratios (RRs) for CES were calculated by dividing the incidence proportion of CES in the CSM cohort by the incidence proportion in the PT evaluation cohort. R (version 4.2.2, Vienna, AT [59]) was used to calculate 95% confidence intervals (CIs) using the normal approximation, and the ggplot2 package [60] was used to plot cumulative incidence (with locally weighted scatterplot smoothing) and propensity score density.

## Study size

Given the lack of previous research to estimate the incidence of individuals with LBP receiving CSM developing CES, a required sample size was estimated using broader epidemiologic data [6]. A required sample size of 103,836 was calculated using G*Power (Kiel University, DE) z-tests for examining a difference in incidence proportion between cohorts (0.3% vs. 0.6% for one year, translating to a possible difference of 0.075% vs. 0.150% per our three-month follow-up) using two tails, an alpha error of 0.05, power of 0.95, and allocation ratio of one. Feasibility testing in February 2023, along with previous work using this dataset [61, 62], suggested that this sample size would be attainable.

## Results

### Participants

Eligible patients were identified from several health care organizations (CSM: 12; PT evaluation: 48). Prior to propensity score matching, the CSM cohort consisted of 67,223 patients, while the PT evaluation cohort consisted of 776,704 patients (Table 1). Following matching, both cohorts consisted of 67,220 patients, with a mean age of 51 years (SD = 17). Before matching, several differences were observed between the CSM cohort and the PT evaluation cohort. Prior to propensity matching, the CSM cohort had a lower mean age, lower proportion of individuals who identified as Asian, Black or African American, and Hispanic or Latino, and lower incidence of several diagnoses and procedures (SMD>0.1 for each). After matching, there were no meaningful differences between the two cohorts for any of the matched variables (SMD <0.1 for each), indicating a successful balancing of the cohorts.

### Descriptive data

The mean number of data points per patient per cohort was adequate (CSM: 3,802; PT evaluation: 5,482). After propensity matching, the proportion of 'unknown' demographic variables was similar between cohorts: unknown race (CSM: 20%, PT evaluation: 19%, SMD = 0.022), unknown ethnicity (both cohorts 16%, SMD = 0.002), unknown sex (both cohorts 0%, SMD<0.001), and unknown age (both cohorts 0%, SMD<0.001). A propensity score density graph indicated that the cohorts were well-balanced after matching (Fig 1). These findings suggested minimal between-cohort differences relating to data completeness, data density, and covariate balance.

### Key results

The incidence of CES over three months' follow-up from the index date of inclusion was lower in the CSM cohort compared to the PT evaluation cohort both before and after matching, yet 95% CIs overlapped (Table 2 and Fig 2). For the primary outcome, after propensity matching, the incidence of CES in the CSM cohort was 0.07% (95% CI: 0.05–0.09%), compared to 0.11% (95% CI: 0.09–0.14%) in the PT evaluation cohort, yielding an RR of 0.60 (95% CI: 0.42–0.86; $p$ = .0052).

### Sensitivity analysis

A cumulative incidence graph demonstrated that the occurrences of CES increased in a curvilinear manner during the first two weeks of follow-up in both cohorts, suggesting a greater risk of CES immediately following the initial visit of CSM or PT evaluation (Fig 3). The 95% CIs for CES incidence overlapped through most of the follow-up window, indicating a similar time trend in both cohorts.

**Table 1. Baseline characteristics before and after propensity score matching.**

| Variable | Before matching | | | After matching | | |
|---|---|---|---|---|---|---|
| | CSM | PT evaluation | SMD | CSM | PT evaluation | SMD |
| | (N = 67,223) | (N = 776,704) | | (N = 67,220) | (N = 67,220) | |
| Mean age (SD) | 50.7(16.9) | 60.1(15.8) | **0.576** | 50.7(16.9) | 50.7(16.8) | 0.002 |
| Age (min-max) | 18–89 | 18–90 | | 18–89 | 18–90 | |
| Female n (%) | 39097 (58%) | 467378 (61%) | 0.052 | 39097 (58%) | 39074 (58%) | 0.001 |
| Male n (%) | 28125 (42%) | 302408 (39%) | 0.052 | 28122 (42%) | 28143 (42%) | 0.001 |
| Body mass index (kg/m$^2$) | 30.2±6.6 | 30.0±7.0 | 0.034 | 30.2±6.6 | 29.9±6.9 | 0.039 |
| Race/ethnicity n (%) | | | | | | |
| Asian | 422 (1%) | 14519 (2%) | **0.113** | 422 (1%) | 452 (1%) | 0.006 |
| Black or African American | 2458 (4%) | 121716 (16%) | **0.419** | 2458 (4%) | 2448 (4%) | 0.001 |
| Hispanic or Latino | 1173 (2%) | 51742 (7%) | **0.249** | 1173 (2%) | 1195 (2%) | 0.002 |
| Not Hispanic or Latino | 55422 (82%) | 585443 (76%) | **0.158** | 55419 (82%) | 55453 (82%) | 0.001 |
| White | 50766 (76%) | 548600 (71%) | 0.096 | 50766 (76%) | 50946 (76%) | 0.006 |
| Diagnosis n (%) | | | | | | |
| Constipation | 2660 (4%) | 107353 (14%) | **0.355** | 2660 (4%) | 2604 (4%) | 0.004 |
| Disc disorder with radiculopathy, lumbar | 602 (1%) | 20232 (3%) | **0.132** | 602 (1%) | 486 (1%) | 0.019 |
| Disc disorder with radiculopathy, lumbosacral | 168 (0%) | 5990 (1%) | 0.074 | 168 (0%) | 124 (0%) | 0.014 |
| Lumbago with sciatica | 2223 (3%) | 59596 (8%) | **0.195** | 2223 (3%) | 2185 (3%) | 0.003 |
| Postlaminectomy syndrome | 341 (1%) | 7279 (1%) | 0.052 | 341 (1%) | 263 (0%) | 0.017 |
| Sciatica | 2772 (4%) | 43868 (6%) | 0.073 | 2771 (4%) | 2674 (4%) | 0.007 |
| Spinal stenosis, lumbar | 2263 (3%) | 74903 (10%) | **0.259** | 2263 (3%) | 1998 (3%) | 0.023 |
| Spondylolisthesis, lumbar | 468 (1%) | 26261 (3%) | **0.192** | 468 (1%) | 469 (1%) | <0.001 |
| Spondylolisthesis, lumbosacral | 180 (0%) | 6981 (1%) | 0.084 | 180 (0%) | 137 (0%) | 0.013 |
| Procedure n (%) | | | | | | |
| Computed tomography, lumbar | 426 (1%) | 31821 (4%) | **0.231** | 426 (1%) | 384 (1%) | 0.008 |
| Emergency department services | 11587 (17%) | 318884 (41%) | **0.551** | 11587 (17%) | 11446 (17%) | 0.006 |
| Epidural steroid injection | 448 (1%) | 8875 (1%) | 0.051 | 448 (1%) | 281 (0%) | 0.034 |
| Magnetic resonance imaging, spinal canal and contents, lumbar | 2941 (4%) | 62653 (8%) | **0.156** | 2941 (4%) | 2753 (4%) | 0.014 |
| Surgical procedures on the spine and spinal cord | 1425 (2%) | 47350 (6%) | **0.204** | 1425 (2%) | 1245 (2%) | 0.019 |
| Surgical procedures on the spine | 193 (0%) | 24623 (3%) | **0.224** | 193 (0%) | 240 (0%) | 0.012 |
| Transforaminal epidural steroid injection | 801 (1%) | 11887 (2%) | 0.030 | 801 (1%) | 596 (1%) | 0.030 |

Abbreviations: Kilograms per square meter (kg/m$^2$), standardized mean difference (SMD), standard deviation (SD), physical therapy (PT)

Bold SMD values (>**0.1**) indicate between-cohort imbalance.

## Discussion

The present study was conducted because prior case reports and medicolegal cases described an onset of CES following CSM [1–4], yet there was no adequately powered and designed study to examine this potential association. The present study tested the hypothesis that there would be no increased risk of CES following CSM, considering limited previous studies suggested this was a rare event and potentially related to pre-existing lumbar disc disorders [1, 8]. The present study results support the hypothesis that there is no increased risk of CES following CSM in adults compared to matched controls receiving PT evaluation without spinal manipulation.

In the present study, the incidence of CES in both cohorts over three months (0.07% to 0.11%) may translate to approximately 0.28% to 0.44% over 12 months, aligning with a previous estimate of CES incidence among individuals with LBP seeking secondary care (0.27%

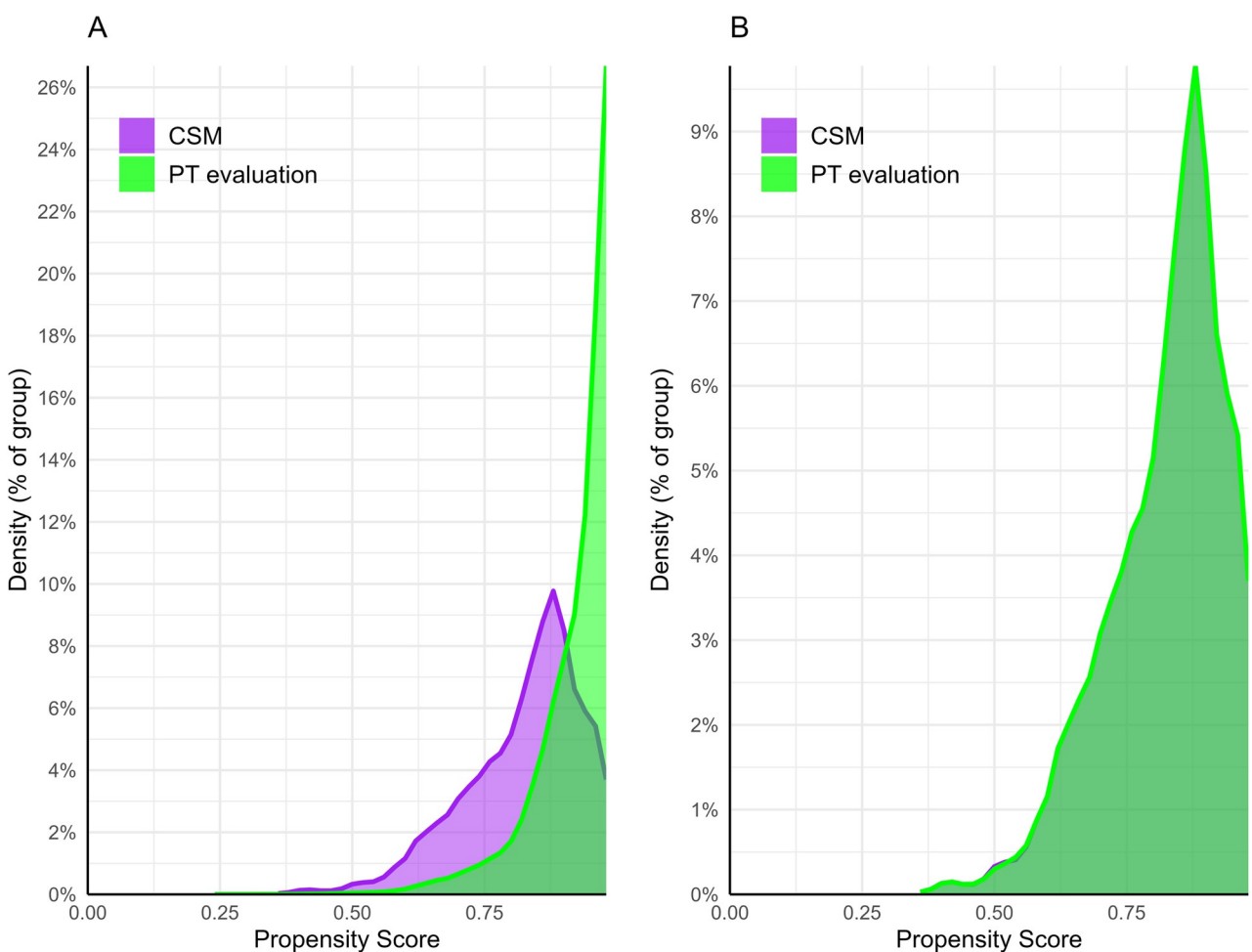

**Fig 1. Propensity score density graph.** Propensity score before (A) and after (B) matching, with purple representing the chiropractic spinal manipulation (CSM) cohort and green representing the physical therapy (PT) evaluation cohort. The area of common support improves after matching, as propensity score densities overlap between cohorts, suggesting adequate balance of covariates.

[95% CI: 0.14–0.54%]) [6]. In the present study, while the RR was significant and less than one, potentially indicating reduced CES risk in the CSM cohort, there was an overlap in the 95% CIs for CES incidence between cohorts, suggesting that any risk difference was not meaningful. The similarity of CES incidence to a prior epidemiologic estimate [6], and similar

**Table 2. Key results before and after propensity score matching.**

|  | Before matching | | After matching* | |
|---|---|---|---|---|
|  | CSM | PT evaluation | CSM | PT evaluation |
|  | (N = 67,223) | (N = 776,704) | (N = 67,220) | (N = 67,220) |
| CES N | 46 | 1,222 | 46 | 77 |
| CES % (95% CI) | 0.07% (0.05–0.09%) | 0.16% (0.15–0.17%) | 0.07% (0.05–0.09%) | 0.11% (0.09–0.14%) |
| RR (95% CI) | 0.44 (0.32–0.58; $p < .0001$) | (reference) | 0.60 (0.42–0.86; $p = .0052$) | (reference) |

Abbreviations: Chiropractic spinal manipulation (CSM), risk ratio (RR), 95% confidence intervals (95% CI), physical therapy (PT)

* primary outcome

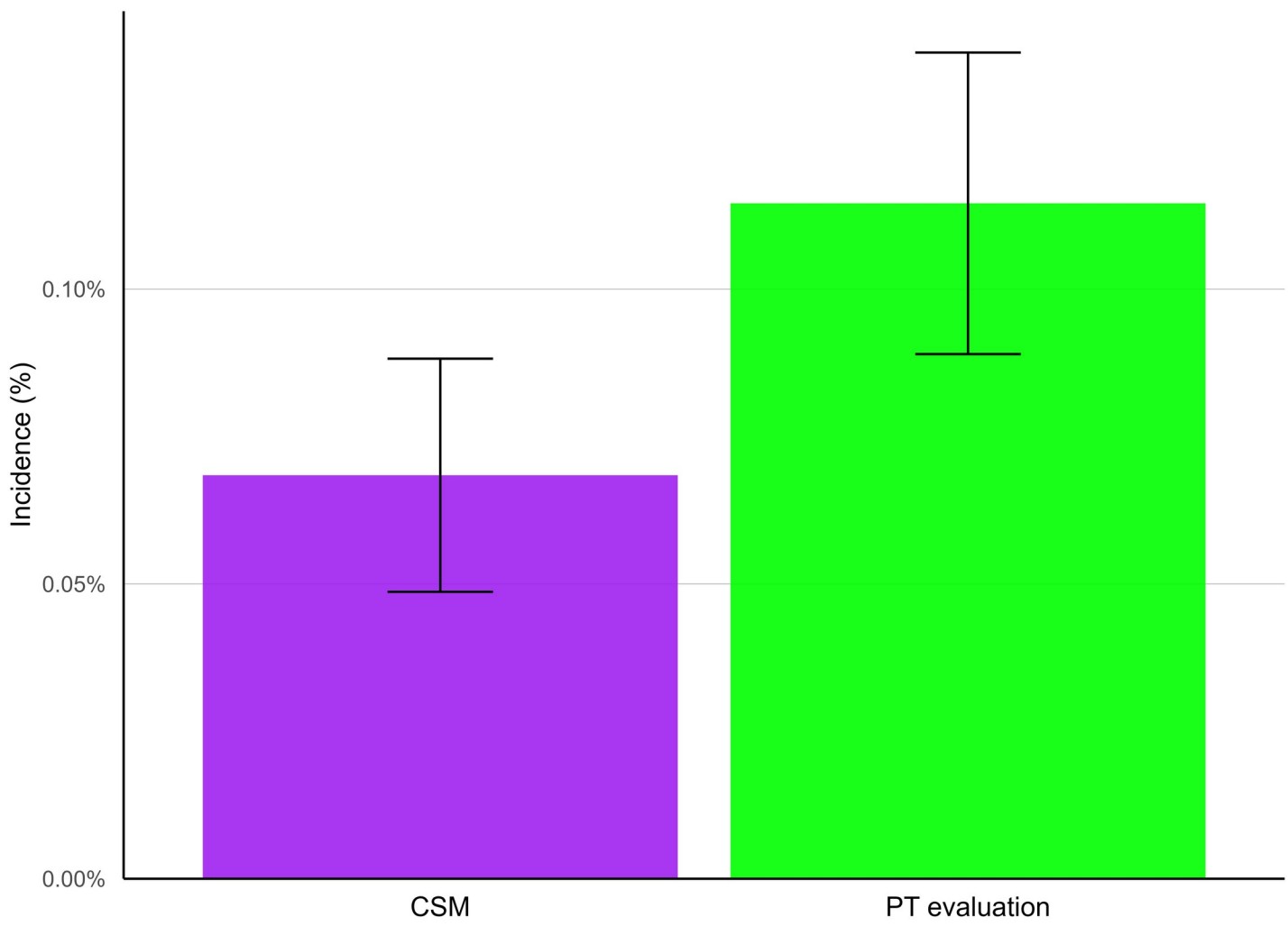

**Fig 2. Incidence of cauda equina syndrome (CES) per cohort after propensity matching.** CES occurs in a smaller proportion of patients in the CSM cohort (purple) compared to the PT evaluation cohort (green), however, the 95% confidence intervals overlap at 0.09%, indicating a potentially non-meaningful difference.

incidence between cohorts, suggest that neither CSM nor PT evaluation influenced the incidence of CES.

A curvilinear increase in CES cumulative incidence in both cohorts was evident over the first two weeks, suggesting the rate of CES is higher when patients first seek care for LBP. This finding aligns with previous evidence that patients with LBP have an inherently higher risk of CES compared to asymptomatic individuals [6]. In addition, this reinforces that clinicians should be vigilant to detect and urgently refer patients with CES symptoms for surgical attention [10].

These present findings contradict the conclusions of prior studies which suggested that an onset of CES after CSM indicated that CSM was causal [12, 63]. However, these prior conclusions were based on case reports, which often highlight atypical situations [64]. In addition, case reports lack a comparator group or a means to account for confounding variables, design elements which were available in the present study.

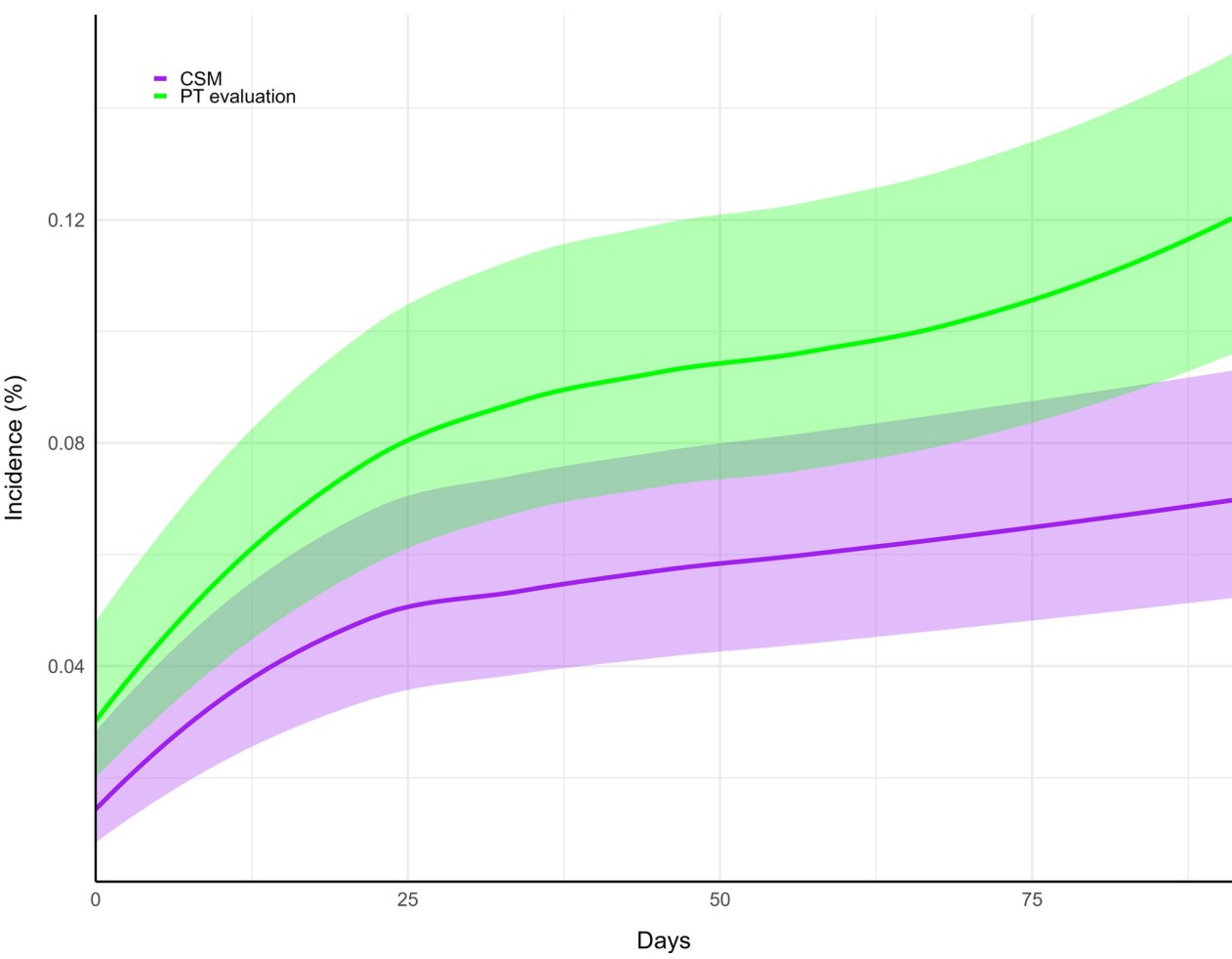

**Fig 3. Cumulative incidence of cauda equina syndrome.** Incidence curves in the chiropractic spinal manipulation cohort (CSM, purple) and physical therapy evaluation cohort (PT evaluation, green) are shown over the three-month follow-up window (90 days). Shaded regions indicate 95% confidence intervals.

Our findings are consistent with the hypothesis that patients who develop CES after CSM may have evolving symptoms of CES prior to treatment and/or an already-existing disc herniation [7, 11]. This phenomenon was illustrated in two medicolegal cases of CES wherein chiropractors appeared to overlook symptoms of acute lumbar disc herniation before administering CSM [7]. The present findings show that CES may also arise soon after PT evaluation without manipulation for LBP, suggesting that patients seeking care for LBP are already at a heightened risk of CES and CSM may not be directly causative.

These findings can be compared to population-based studies examining the association between CSM and lumbar disc herniation. A self-controlled case series found that patients who underwent emergency surgery for acute lumbar disc herniation had a similar increase in likelihood of visiting either primary care providers or chiropractors prior to the surgery, suggesting that CSM was not a risk factor for lumbar disc herniation [13]. Another study found that patients with radicular LBP who underwent CSM were less likely to require disc surgery over the subsequent two years compared to matched controls receiving usual medical care [20]. These consistent findings support the notion that CSM is not a meaningful risk factor for disc herniation or CES.

There are multiple reasons why one may suspect that CSM would not contribute to CES. Biomechanically, the lumbar facet joints limit axial rotation during manipulation, thereby protecting the lumbar intervertebral discs [14]. As CSM includes a range of approaches (e.g., high- or low-velocity or force and manual or instrument-assisted manipulations [65]), chiropractors' ability to customize CSM to patients' clinical presentations could reduce the likelihood of adverse outcomes. Speculatively, chiropractors could avoid higher-force manipulations on patients with more severe LBP presentations, potentially reducing risk of harm.

Alternate study designs could be considered to corroborate the present findings. A self-controlled case series would use patients as their own controls, minimizing biases with respect to clinical features. A case-control design could be used to compare CES cases with matched non-CES controls and allow for the examination of a potential dose-response relationship between CSM and CES. Alternate comparator groups could be examined, considering different clinician types (e.g., primary care, orthopedic surgeons) may encounter LBP patients of varying complexity and baseline CES risk. Detailed cases from chart reviews, registries, or databases are also needed to understand the clinical presentation and steps to CES diagnosis among patients presenting to chiropractors with LBP.

## Strengths and limitations

Strengths of this study include the use of a large, multicenter sample of over 130,000 total patients, a multidisciplinary research team, propensity matching strategy, and use of a registered protocol [32]. The observational design of this study precludes any definitive conclusion regarding the potential for a causal relationship between CSM or PT evaluation and CES, or a lack thereof. Residual confounding related to unmeasured risk factors such as LBP-related disability may have influenced our results. Subtypes of LBP diagnoses may have been incorrect in the medical record due to our lack of requiring previous diagnostic imaging or testing; for example, a label of sciatica includes a broad differential diagnosis encompassing neuropathy and other conditions. Due to the de-identified nature of the dataset, it was not possible to validate CES outcomes via chart review, and false positives or missed diagnoses are possible. Limitations in data granularity prevented the description of parameters of CSM application that may be relevant to CES risk, such as force and type of thrust [65]. The years of chiropractors' experience and any additional post-graduate training was not feasible to examine in the current study, which could play a role in risk mitigation with SMT. Study results may not be broadly generalizable as treatment of LBP and diagnosis of CES may vary differ in smaller private practice settings as well as regionally with respect to chiropractic and PT approaches outside of the US. In addition, these findings pertain to spinal manipulation administered by trained chiropractors rather than other practitioners or laypersons, considering cases of severe adverse events including spinal fracture and CES have been reported following spinal manipulation by untrained individuals [45, 66–68].

## Conclusions

The present study involving over 130,000 propensity-matched patients found that CSM is not a risk factor for CES. The incidence of CES in both CSM and PT evaluation cohorts aligns with previous estimates of CES incidence among patients with LBP, indicating a heightened risk of CES compared to asymptomatic individuals regardless of intervention. Moreover, these findings underscore the increased CES incidence within the first two weeks after either CSM or PT evaluation, emphasizing the need for clinicians' vigilance in identifying and emergently referring patients with CES for surgical evaluation. Further real-world evidence is needed to

corroborate these findings using alternative case-control and case-crossover designs, and different clinician comparators.

## Supporting information

**S1 Fig. Study design.**
(DOCX)

**S1 Table. Inclusion codes for both cohorts.**
(DOCX)

**S2 Table. Exclusion codes for both cohorts.**
(DOCX)

**S3 Table. Variables controlled for in propensity score matching.**
(DOCX)

## Author Contributions

**Conceptualization:** Robert J. Trager, Anthony N. Baumann, Jeffery A. Dusek, Christine M. Goertz.

**Data curation:** Robert J. Trager, Jaime A. Perez.

**Formal analysis:** Robert J. Trager, Jaime A. Perez.

**Investigation:** Robert J. Trager, Anthony N. Baumann, Jaime A. Perez, Christine M. Goertz.

**Methodology:** Robert J. Trager, Anthony N. Baumann, Jaime A. Perez, Christine M. Goertz.

**Project administration:** Jeffery A. Dusek, Christine M. Goertz.

**Resources:** Jaime A. Perez.

**Software:** Robert J. Trager, Jaime A. Perez, Romeo-Paolo T. Perfecto.

**Supervision:** Jeffery A. Dusek, Christine M. Goertz.

**Visualization:** Robert J. Trager, Jaime A. Perez, Romeo-Paolo T. Perfecto.

**Writing – original draft:** Robert J. Trager.

**Writing – review & editing:** Robert J. Trager, Anthony N. Baumann, Jaime A. Perez, Jeffery A. Dusek, Romeo-Paolo T. Perfecto, Christine M. Goertz.

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
