## [Decision Letter · Decision Letter 0]

25 Jan 2024

PONE-D-23-39784Association between chiropractic spinal manipulation and cauda equina syndrome in adults with low back pain: Retrospective cohort study of US academic health centersPLOS ONE

Dear Dr. Trager,

Thank you for submitting your manuscript to PLOS ONE. After careful consideration, we feel that it has merit but does not fully meet PLOS ONE’s publication criteria as it currently stands. Therefore, we invite you to submit a revised version of the manuscript that addresses the points raised during the review process.

We look forward to receiving your revised manuscript.

Kind regards,

Shabnam ShahAli, Ph.D.

Academic Editor

PLOS ONE

Journal Requirements:

4. Thank you for stating the following in the Acknowledgments Section of your manuscript: "This publication was made possible through the support of the Clinical Research Center of University Hospitals Cleveland Medical Center (UHCMC) and the Case Western Reserve University Clinical and Translational Science Collaborative (CTSC) 4UL1TR000439. Its contents are solely the responsibility of the authors and do not necessarily represent the official views of UHCMC or National Institutes of Health."

Please remove any funding-related text from the manuscript and let us know how you would like to update your Funding Statement. Currently, your Funding Statement reads as follows: "The authors received no specific funding for this work."

Reviewers' comments:

Reviewer's Responses to Questions

**Comments to the Author**

1. Is the manuscript technically sound, and do the data support the conclusions?

Reviewer #1: Yes

Reviewer #2: Yes

Reviewer #3: Yes

Reviewer #4: Yes

Reviewer #5: Yes

2. Has the statistical analysis been performed appropriately and rigorously? 

Reviewer #1: Yes

Reviewer #2: Yes

Reviewer #3: Yes

Reviewer #4: Yes

Reviewer #5: Yes

3. Have the authors made all data underlying the findings in their manuscript fully available?

Reviewer #1: Yes

Reviewer #2: Yes

Reviewer #3: Yes

Reviewer #4: Yes

Reviewer #5: Yes

4. Is the manuscript presented in an intelligible fashion and written in standard English?

Reviewer #1: Yes

Reviewer #2: Yes

Reviewer #3: Yes

Reviewer #4: Yes

Reviewer #5: Yes

5. Review Comments to the Author

Reviewer #1: Please note and correct the following:

1. The words like “we” and “our” were used a lot in this study. It is not common to use such words in scientific articles. It is better to rewrite the sentences in the passive form. Please edit this in the entire text.

2. In the abstract section, the division of patients into two cohort groups is not clear for the reader. Please explain it clearly. If this is what you mean, explain it like this; LBP patients who received manipulation by chiropractors and LBP patients who underwent evaluation and treatment without manipulation by physical therapists.

3. In key word section, is it necessary to use “safety” as a key word? This could show that your search was done with a positivity bias.

4. In the last paragraph of the introduction section, the use of the word “crucial” is not appropriate and again shows a positivity bias towards spinal manipulation.

5. The phrase “Incidence of CES per cohort” is unnecessary and should be removed from the top of the figure 1.

6. The primary outcome was not mentioned in table 2, what is the purpose of the sign * for? It seems better to mention it in the title of the table for example: “score matching for primary outcomes). In addition, it seems that the sign * along with its explanation should be omitted from the table footer 1.

7. Please match the references to the relevant sentences. For example: A) In the third paragraph of the introduction section, it is not possible that the results of article 11 (a systematic review article) is referenced for the two statements that CSM can be a risk factor for CES or not. B) reference 31 seems not to be related to the last sentence of limitation section.

8. The discussion is very poorly written. Logical and scientific justifications were not mentioned for the hypothesis. Please write strong and logical justifications for your hypothesis. Note that the way the discussion is written can be a reason for rejecting or accepting an article.

9. In the discussion section, the sentence “Limitations of such cases include their small sample size and potential confounding factors” is incomprehensible, please explain it clearly.

10. In the limitation section, the sentence “Further research is needed to replicate our findings and assess their consistency” is only a suggestion for future research and not a limitation. Please omit this from this section.

11. In the reference list, the names of the journals should be written in an abbreviated form.

Reviewer #2: The area of investigation has chosen wisely as manipulation is gaining adherents day by day. The manuscript is well written and technically sound. There is only one question from author, how did you include or exclude diabetic patients with neuropathy? please explain your examination and diagnostic tests for confirmation of radicular or sciatic pain.

Reviewer #3: How do you consider 90-day follow-up?

It is important for this study that expert or novice chiropractic does thrust manipulation. Why do you include years of experience of chiropractic?

Method of Abstract: it doesn’t need deception exclusion criteria in abstract

Reviewer #4: Dear authors, I am very happy to be able to write positively. It is evident that you have put a great deal of effort into this project and I want to praise your efforts, Fortunately, the actual contribution from your study is clear and, the manuscript as currently written suggests that it might be suitable for sharing information about this topic, but the manuscript that you reported, needs few minor edits.

General comment: Authors must consider using line numbers in manuscripts.

Abstract

Line: ‘following physical therapy (PT) evaluation’: It can be better written as ‘intervention’, instead of ‘evaluation’.

‘Patients were divided into two cohorts: (1) CSM or (2) PT evaluation without spinal Manipulation’: Did the patients underwent one time evaluation (assessment) or were they on specific physiotherapeutic interventions for any specific time (such as a week, or two)? If they were on intervention(s), the word ‘evaluation’ should be replaced with the word ‘intervention’.

‘Propensity score matching controlled for covariates associated with CES (e.g., body mass index).’: correct the syntax of the sentence.

Lines ‘(mean age 51 years [SD=17]) remained after propensity matching. CES incidence was 0.07% (95% CI: 0.05-0.09%) in the CSM cohort compared to 0.11% (95% CI: 0.09- 0.14%) in the PT evaluation cohort, yielding an RR (95% CI) of 0.60 (0.42-0.86; P=.0052). Both cohorts showed a higher rate of CES during the first two weeks of follow-up.’: Elaborate the abbreviations at first notice.

Introduction: No comments

Materials and Methods

Rephrase the line ‘We included patients starting 20 years ‘prior to the query date (July 30, 2023) to maximize sample size’ for a clearer understanding.

Results

Table 1: Add n (%) for males and females to let the readers understand that the values are in terms of n and its corresponding percentage. Also, add SI units for BMI (kg/m2) and other such variables in the tables.

The level of significance (p) should be written in small italicized letters.

Discussion No comments

Limitations & Conclusions: No comments

Reviewer #5: This study investigated association between chiropractic spinal manipulation and cauda equina syndrome in adults with low back pain. The authors concluded that CSM is not a risk factor for CES. Considering prior epidemiologic evidence, patients with LBP may have an elevated risk of CES independent of treatment. These findings warrant further corroboration. In the meantime, clinicians should be vigilant to identify LBP patients with CES and promptly refer them for surgical evaluation.

Comment#1

The authors should insert low back pain as a keywords.

Comment#2

The authors should provide more explanations about the risk of Cauda equina syndrome following chiropractic spinal manipulation and physiotherapy.

Comment#3

The authors should state more explanations for performing your study.

6. PLOS authors have the option to publish the peer review history of their article (what does this mean?). If published, this will include your full peer review and any attached files.

Reviewer #1: No

Reviewer #2: **Yes: **Ghazal Kharaji

Reviewer #3: No

Reviewer #4: No

Reviewer #5: No

---

## [Author Response · Author response to Decision Letter 0]

30 Jan 2024

Please see the response document file titled "Response to reviewers" towards the end of the review file PDF - which may be formatted better, although our responses are also copied here:

Overview

We are grateful for the thorough and detailed reviewer comments and have made several revisions to strengthen our manuscript. We have added more detailed justification for conducting the study, focused the discussion on key elements that align with our hypothesis, changed wording to the passive voice, removed unnecessary material from the text and tables, and made formatting and reference style changes, among other edits. 

Editorial comments

• Response: Thank you. See changes:

o We apologize that we were not using the journal abbreviations in the references – which we fixed. They are already in Vancouver style. 

o We added continuous line numbers

o We removed the 4th level headings, now we only use level 1 and level 2 headings

o We added SI units for body mass index

o We altered the font sizes, including the heading font sizes, to match the documents you sent. 

o Supporting information is already labeled according to S1, S2 etc. 

o Please let us know if there is anything we need to do to adhere to the style requirements.

• Response: Thank you. We checked to ensure that our data was shared in a way that matched journal requirements, using an open access repository with an available doi and link. Please let us know if there is anything we need to do to improve this. We inform readers how to access the data in the Data availability statement.

• Response: Thank you. Our study did not include minors, as it was limited to adults (age 18 or older). We updated our statement to explicitly state that the need for consent was waived in the methods section, and mention that data were fully anonymized. 

• This study used fully anonymized, de-identified data and therefore was deemed Not Human Subjects Research by the University Hospitals Institutional Review Board (Cleveland, Ohio, USA, STUDY20230269), which also waived the need for consent. TriNetX is compliant with the Health Insurance Portability and Accountability Act (HIPAA) [1]. TriNetX only contains de-identified data as per the de-identification standard defined in Section §164.514(a) of the HIPAA Privacy Rule. The TriNetX network contains data provided by participating healthcare organizations, each of which represents and warrants that it has all necessary rights, consents, approvals, and authority to provide the data to TriNetX under a Business Associate Agreement, so long as their name remains anonymous as a data source and their data are utilized for research purposes. The data shared through the TriNetX Platform are attenuated to ensure that they do not include sufficient information to facilitate the determination of which health care organization contributed which specific information about a patient.

• Response: Thank you. Please see the updated statement above which mentions these components. Also note that we have an IRB approval / exemption document that we can share with the Editor upon request.

4. Thank you for stating the following in the Acknowledgments Section of your manuscript: "This publication was made possible through the support of the Clinical Research Center of University Hospitals Cleveland Medical Center (UHCMC) and the Case Western Reserve University Clinical and Translational Science Collaborative (CTSC) 4UL1TR000439. Its contents are solely the responsibility of the authors and do not necessarily represent the official views of UHCMC or National Institutes of Health."

Please remove any funding-related text from the manuscript and let us know how you would like to update your Funding Statement. Currently, your Funding Statement reads as follows: "The authors received no specific funding for this work."

• Response: Thank you. We deleted the previous funding-related text from the acknowledgement section and added it to the funding section. We also added this statement to the online submission form – replacing our previous statement. Please note that the funding statement is also updated and improved as follows:

o This project was supported by the Clinical and Translational Science Collaborative of Northern Ohio which is funded by the National Institutes of Health, National Center for Advancing Translational Sciences, Clinical and Translational Science Award grant, UM1TR004528. The content is solely the responsibility of the authors and does not necessarily represent the official views of the National Institutes of Health.

• Response: Thank you. We also inserted the funding statement in the revised cover letter.

• Response: Thank you. We reviewed the reference list and changed a setting so that it now lists the journal abbreviations as recommended. We did not see any retracted articles – and we are using Zotero which has a built-in feature in collaboration with Retractionwatch to alert us if there is any retraction in the cited studies.

Reviewer #1

Please note and correct the following:

1. The words like “we” and “our” were used a lot in this study. It is not common to use such words in scientific articles. It is better to rewrite the sentences in the passive form. Please edit this in the entire text.

• Response: Thank you for this suggestion. We edited the text throughout to remove these terms and now use the passive voice.

2. In the abstract section, the division of patients into two cohort groups is not clear for the reader. Please explain it clearly. If this is what you mean, explain it like this; LBP patients who received manipulation by chiropractors and LBP patients who underwent evaluation and treatment without manipulation by physical therapists.

• Response: Thank you for pointing out that this description of the cohorts could be clearer. We wish to note that earlier in the abstract “chiropractic spinal manipulation” was already abbreviated as “CSM” and “physical therapy” was abbreviated as “PT” so we can reuse these acronyms here. Also, please note that we did not necessarily require any specific treatment in the PT cohort – simply an evaluation. We took your suggestion and revised this sentence as follows:

o Patients were divided into two cohorts: (1) LBP patients receiving CSM or (2) LBP patients receiving PT evaluation without spinal manipulation.

3. In key word section, is it necessary to use “safety” as a key word? This could show that your search was done with a positivity bias.

• Response: Thank you for this recommendation. We have removed the term “safety” from the keywords. We agree this might give the appearance of a bias. However, please note that we took several steps to minimize bias including having a multidisciplinary team and registering our protocol. We also now do a better job of emphasizing potential risks of CSM in the introduction (see comments for Reviewer #5) to show a better balance in our writing.

4. In the last paragraph of the introduction section, the use of the word “crucial” is not appropriate and again shows a positivity bias towards spinal manipulation.

• Response: Thank you, we agree this term could be changed. We removed the term “crucial” as advised, changing it to a more neutral term of “necessary”. However, please note that this sentence is referring to the possible risks of spinal manipulation from the prior literature and justifying why our study is necessary - to examine potential risks further. 

5. The phrase “Incidence of CES per cohort” is unnecessary and should be removed from the top of the figure 1.

• Response. Thank you for this suggestion, we agree completely and feel that removing the title better highlights the data. We believe you mean Figure 2, as there is no title on Figure 1. We removed the title “Incidence of CES per cohort” from the Figure 2 plot area

6. The primary outcome was not mentioned in table 2, what is the purpose of the sign * for? It seems better to mention it in the title of the table for example: “score matching for primary outcomes). In addition, it seems that the sign * along with its explanation should be omitted from the table footer 1.

• Response: Thank you, we agree that this is confusing and unnecessary. We removed the asterisk. The propensity matched cohorts were also used for all outcomes, including our sensitivity analysis so we feel we should keep the title the same, and not try to emphasize which were primary outcomes in this table. The purpose of matching is explained in the Methods section.

7. Please match the references to the relevant sentences. For example: A) In the third paragraph of the introduction section, it is not possible that the results of article 11 (a systematic review article) is referenced for the two statements that CSM can be a risk factor for CES or not. B) reference 31 seems not to be related to the last sentence of limitation section.

• Response: Thank you – we fixed the first potential reference issue as advised by adding a different references supporting spinal manipulation as a potential risk factor for cauda equina syndrome instead of the review article (previously #11 [2]). However, please note that that the original reference was a review of legal / malpractice cases and the author presented both sides to the argument, that CSM might be causative, and it might not be causative of CES. Regardless, the new references may be better for readers [3,4]. The new references are as follows:

o Haldeman, Scott, and Sidney M. Rubinstein. "Cauda equina syndrome in patients undergoing manipulation of the lumbar spine." Spine 17.12 (1992): 1469-1473.

o Tamburrelli, Francesco Ciro, Maurizio Genitiempo, and Carlo Ambrogio Logroscino. "Cauda equina syndrome and spine manipulation: case report and review of the literature." European Spine Journal 20 (2011): 128-131.

• We edited the statement in the limitations section to better match the cited reference and added new references as well. The original sole reference (Hebert et al) review article notes several adverse events such as cauda equina syndrome that occurred after spinal manipulation. Several of the treating clinicians were not chiropractors or they had no known or reported credentials. We are suggesting that our results showing the lack of heightened risk of CES after CSM do not apply to manipulation performed by untrained individuals. Essentially, manipulation may be more dangerous with a lack of training:

o In addition, these findings pertain to spinal manipulation administered by trained chiropractors rather than other practitioners or laypersons, considering cases of severe adverse events including spinal fracture and CES have been reported following manipulation by untrained individuals [5–8].

o Please also note these references are also added to support the original one:

Yang, Si-Dong, Qian Chen, and Wen-Yuan Ding. "Cauda equina syndrome due to vigorous back massage with spinal manipulation in a patient with pre-existing lumbar disc herniation: a case report and literature review." American Journal of Physical Medicine & Rehabilitation 97.4 (2018): e23-e26.

Terrett, A. G. "Misuse of the literature by medical authors in discussing spinal manipulative therapy injury." Journal of manipulative and physiological therapeutics 18.4 (1995): 203-210.

Wenban, Adrian B. "Inappropriate use of the title'chiropractor'and term'chiropractic manipulation'in the peer-reviewed biomedical literature." Chiropractic & Osteopathy 14.1 (2006): 1-7.

8. The discussion is very poorly written. Logical and scientific justifications were not mentioned for the hypothesis. Please write strong and logical justifications for your hypothesis. Note that the way the discussion is written can be a reason for rejecting or accepting an article.

• Response. Thank you. We revised the Discussion entirely and are grateful for the opportunity to improve this section. We now realize that several sentences lacked a clear connection to the key points of the study. Our re-write strived for clarity and communicating to a broader audience, while remaining focused and relevant to the main study objectives.

• We trimmed or deleted several sentences and paragraphs to make the Discussion more concise, added a brief recap of the justification for performing the study as advised, reorganized some information, changed the “Limitations” section to “Strengths and limitations” for better organization, and removed some jargon and confusing phrases. 

• Please note that while we hypothesized there would be no increased risk of CES after CSM, we were aware (based on our feasibility testing) that we were working with a large sample size, therefore we implemented a 2-tailed alpha in our sample size calculation, ensuring we were adequately powered to test any direction of association (positive, null, or negative). Had there been a positive association between CSM and CES, our sample size and methods would have enabled us to identify this outcome.

• Please see the newly added beginning sentence to the Discussion, which begins with a logical scientific justification for our hypothesis, while you may see several other edits in the revised draft:

o The present study was conducted because prior case reports and medicolegal cases described an onset of CES following CSM [2–5], yet there was no adequately powered and designed study to examine this potential association. The present study tested the hypothesis that there would be no increased risk of CES following CSM, considering limited previous studies suggested this was a rare event and potentially related to pre-existing lumbar disc disorders [2,9]. The present study results support the hypothesis that there is no increased risk of CES following CSM in adults compared to matched controls receiving PT evaluation without spinal manipulation.

• Please note that Reviewer #5 asked for us to add more background information to help justify the study, and we have added much more detail to the Introduction section which highlights the increase in use of CSM for LBP in the US, potential for harm when treating a population with LBP and potentially at risk for CES, and the limited available evidence on the topic. These changes may also help clarify our justification and rationale for conducting the study.

9. In the discussion section, the sentence “Limitations of such cases include their small sample size and potential confounding factors” is incomprehensible, please explain it clearly.

• Response: Thank you for pointing out that this was unclear. We have re-written this entire paragraph for clarity. See below:

o These present findings contradict the conclusions of prior studies which suggested that an onset of CES after CSM indicated that CSM was causal [4,10]. However, these prior conclusions were based on case reports, which often highlight atypical situations [11]. In addition, case reports lack a comparator group or a means to account for confounding variables, design elements which were available in the present study. 

10. In the limitation section, the sentence “Further research is needed to replicate our findings and assess their consistency” is only a suggestion for future research and not a limitation. Please omit this from this section.

• Response: Thank you. We agree and we have removed this sentence from the limitations section.

11. In the reference list, the names of the journals should be written in an abbreviated form.

• Response: Thank you. We modified these accordingly to use journal abbreviations.

Reviewer #2: 

The area of investigation has chosen wisely as manipulation is gaining adherents day by day. The manuscript is well written and technically sound. There is only one question from author, how did you include or exclude diabetic patients with neuropathy? please explain your examination and diagnostic tests for confirmation of radicular or sciatic pain.

• Response: Thank you for the positive comments. Note that we required patients to have only one of several diagnosis codes for low back pain, and they were not necessarily required to have radicular or sciatic pain. For example, see in Table 2 that only 4% of each cohort had sciatica on initial presentation. Our broad inclusion of several types of low back pain was done to increase sample size, yet we propensity matched so that cohorts would have a similar composition of each type of LBP.

• We acknowledge that we did not require patients to have any specific diagnostic testing, such as MRI, EMG, NCV, etc. While this may have been helpful, this would have narrowed the cohort sizes as many types of low back pain are diagnosed clinically without any imaging. Our study population generally focuses on patients that are seeking care and therefore may not have had many prior diagnostic tests, as opposed to those who have undergone extensive care already. Note in Table 2 that only 4% of patients per cohort had a prior lumbar spine MRI at presentation, 1% had a CT scan, and at most, 1% had a lumbar epidural steroid injection. The low rate of prior imaging is also expected given the American College of Radiology criteria for imaging for low back pain [12]. 

• Our study relied on several methods to help define a cohort of patients with low back pain. Aside from inclusion criteria, we excluded patients with several comorbidities including bladder dysfunction, fecal incontinence, and serious spinal pathology who may have already had CES or syndromes that resembled it. In addition, we used natural language processing to further strengthen our selection criteria, using software which has been validated previously. This process was used to refine the query that we used which was mostly based on ICD-10 codes. However, we realize we should have described the natural language processing better because it helps to show a strength of our approach. See the added sentences:

Prior studies have demonstrated that this software has acceptable accuracy, reliability, and agreement when compared to manual chart review for extracting clinical concepts related to diagnoses, laboratory values, medications, and symptoms [13,14].

o To our knowledge, diabetic neuropathy is not a risk factor for cauda equina syndrome [15–20], although we did control for body mass index, which is a known risk factor .

o However, we agree that it is possible patients had an incorrect diagnosis of one of the subtypes of low back pain that we included, and added a limitation about this:

Subtypes of LBP diagnoses may have been incorrect in the medical record due to our lack of requiring previous diagnostic imaging or testing; for example, a label of sciatica includes a broad differential diagnosis encompassing neuropathy and other conditions.

Reviewer #3: 

How do you consider 90-day follow-up?

• Response: Thank you for this question. In the Methods section we state that “identification of occurrences of CES was over a 90-day follow-up window commencing from the index date of CSM or PT evaluation”. The cumulative incidence graph also shows the days on the X-axis (Figure 3)

• Note that we chose a relatively long 90-day follow-up window because CES did not necessarily develop immediately in prior reports of this condition following CSM, and CES may take time to be identified/diagnosed [21–23]. While much of the literature focuses on this condition being an emergency and recommends prompt surgery, real-world data suggest that it may take up to 90 days to diagnose this condition [23]. Another reason is that we did not want to miss any cases of CES by ending the follow-up window too soon. We also felt that our cumulative incidence graph helps show CES occurrences throughout the 90-day window and could help characterize whether any potential risk window was early, intermediate, or late during this period.

• Please let us know if there is anything you would like us to improve regarding this information. 

It is important for this study that expert or novice chiropractic does thrust manipulation. 

• Response: Thank you. We added a general disclaimer that chiropractors employed in the study settings typically have multiple years of clinical experience (see our response to your next comment below). We also wish to point out that practicing chiropractic in the US requires a doctoral degree and board examination, which suggests that there is some baseline standardization. However, we agree with your point that the level of experience may matter. We added a limitation as follows: 

o The years of chiropractors’ experience and any additional post-graduate training was not feasible to examine in the current study, which could play a role in risk mitigation with SMT.

Why do you include years of experience of chiropractic?

• Response: Thank you for this question. We did not include this data because it was not available in the dataset for which we have access. We agree that years of chiropractic experience could be relevant when considering the likelihood of adverse events such as CES. For example, chiropractors who have more experience might better recognize patients with CES and acknowledge when to refer them for emergency care rather than use spinal manipulation. However, there is not sufficient evidence to substantiate this idea in the context of chiropractic and CES since we are currently conducting the largest study on the topic. 

• We edited the Methods section to add more detail regarding potential years of experience and inform readers that the practice setting is relatively unique considering the healthcare organizations included in our study are hospitals and academic institutions. We also should have also noted that there is a standard educational requirement for chiropractic clinicians in the US. Please see the improved section below regarding education, scope of practice, and years of experience:

o Precise data regarding the characteristics of chiropractors and PTs in the included healthcare organizations (e.g., years of experience, additional training) was not available due to de-identification of the dataset. In general, US chiropractors must complete a doctoral-level degree and pass the National Chiropractic Board of Chiropractic Examiners examinations [24]. In addition, the chiropractic scope of practice is legally regulated [25], and each US state requires continuing education credits [26]. However, evidence suggests that only a minority of chiropractic and PT clinicians in the US are employed in a hospital-based practice setting such as those included in the TriNetX dataset [27,28]. One study reported that chiropractors in hospital-based settings were a mean 21 years’ post-graduation (minimum: 2 to maximum: 40) [29].

Method of Abstract: it doesn’t need deception exclusion criteria in abstract

• Response: Thank you. We deleted the specific exclusion criteria examples such as malignancy, fracture, and infection, from the abstract as we agree this phrase was too detailed.

Reviewer #4: 

Dear authors, I am very happy to be able to write positively. It is evident that you have put a great deal of effort into this project and I want to praise your efforts, Fortunately, the actual contribution from your study is clear and, the manuscript as currently written suggests that it might be suitable for sharing information about this topic, but the manuscript that you reported, needs few minor edits.

General comment: Authors must consider using line numbers in manuscripts.

• Response: Thank you. We have added line numbers as suggested in the revised document.

Abstract

Line: ‘following physical therapy (PT) evaluation’: It can be better written as ‘intervention’, instead of ‘evaluation’.

‘Patients were divided into two cohorts: (1) CSM or (2) PT evaluation without spinal Manipulation’: Did the patients underwent one time evaluation (assessment) or were they on specific physiotherapeutic interventions for any specific time (such as a week, or two)? If they were on intervention(s), the word ‘evaluation’ should be replaced with the word ‘intervention’.

• Response: Thank you for this comment. We suspect this term was not clear. Please note that we only required patients who saw a physical therapist (PT) to have an evaluation (e.g., an examination) and we did not require them to have any treatment. We only required the PT patients to not have anything like spinal manipulation or manual therapy which could resemble the chiropractic treatments in the other cohort. The study is based on the receipt of CSM or PT evaluation on the index date of inclusion and we did not track the type of care that the patients received afterwards, during follow-up. We agree that this could have been interesting to track, yet it was beyond the scope of our study. We were most concerned with identifying CES during the follow-up window, rather than characterizing the types of interventions patients received after their initial visit. 

• Note that we used the term “evaluation” to be consistent with the definition of the procedural codes that we used in the PT cohort. These are defined using the term “evaluation” so we were hesitant to change it to a similar term or synonym such as “examination” (see example here: https://www.aapc.com/codes/cpt-codes/97161)

‘Propensity score matching controlled for covariates associated with CES (e.g., body mass index).’: correct the syntax of the sentence.

• Response: Thank you. We deleted the text describing this example “(e.g., body mass index)”. We used propensity matching for several variables and it is challenging to try to list them all in the abstract. Similar to your comment above about listing examples of exclusions, we felt it was too detailed to try to include examples of matched variables in the abstract. In exchange, we replaced “covariates” with “confounding variables” for clarity.

Lines ‘(mean age 51 years [SD=17]) remained after propensity matching. CES incidence was 0.07% (95% CI: 0.05-0.09%) in the CSM cohort compared to 0.11% (95% CI: 0.09- 0.14%) in the PT evaluation cohort, yielding an RR (95% CI) of 0.60 (0.42-0.86; P=.0052). Both cohorts showed a higher rate of CES during the first two weeks of follow-up.’: Elaborate the abbreviations at first notice.

• Response: Thank you, we are grateful that you noticed this issue. 

o We elaborated on the abbreviation confidence intervals (CI). 

o We removed the abbreviation “RR” and now state “risk ratio” because we only used this abbreviation once in the abstract. It is defined again later in the main text.

o We removed the abbreviation SD because it was challenging to fit “standard deviation (SD)” into this section while maintaining a readable style and meeting the word limit requirements. We also feel that the SD for age is not necessary to report in the abstract as it was not a main outcome.

Materials and Methods

Rephrase the line ‘We included patients starting 20 years ‘prior to the query date (July 30, 2023) to maximize sample size’ for a clearer understanding.

• Response: Thank you. We added the apostrophe on years as requested.

Results

Table 1: Add n (%) for males and females to let the readers understand that the values are in terms of n and its corresponding percentage. Also, add SI units for BMI (kg/m2) and other such variables in the tables.

• Response: Thank you. We added the nomenclature that you kindly provided to Table 1. We feel this greatly helps with interpretation.

The level of significance (p) should be written in small italicized letters.

• Response: Thank you. We have made this change as advised throughout the manuscript to write “p”

Reviewer #5

This study investigated association between chiropractic spinal manipulation and cauda equina syndrome in adults with low back pain. The authors concluded that CSM is not a risk factor for CES. Considering prior epidemiologic evidence, patients with LBP may have an elevated risk of CES independent of treatment. These findings warrant further corroboration. In the meantime, clinicians should be vigilant to identify LBP patients with CES and promptly refer them for surgical evaluation.

Comment#1

The authors should insert low back pain as a keywords.

• Response: Thank you. We added “low back pain” as a keyword.

Comment#2

The authors should provide more explanations about the risk of Cauda equina syndrome following chiropractic spinal manipulation and physiotherapy.

• Response: Thank you. We added more information in the introduction section as advised which responds to this comment and the comment below. See the added material:

o Chiropractors are increasingly sought by patients in the US for the treatment of LBP [30]. A recent study based on insurance claims revealed that chiropractors were among the most commonly visited healthcare providers for new episodes of LBP, ranking second only to primary care physicians (25.2% of episodes with primary care versus 24.8% with a chiropractor) [31]. Furthermore, chiropractors use spinal manipulation more frequently than any other type of clinician [31]. 

o Half of chiropractic patients have LBP, [32] with a subset of these patients having lumbar disc herniation [33]. Although CES is a rare event, lumbar disc herniation is its most common cause [34] and is also frequently present among those with LBP [35]. Accordingly, chiropractors may encounter patients who have a heightened risk of developing CES, as these clinicians treat those with LBP and disc disorders [3,9]. 

o Considering CSM is commonly used for LBP, wherein underlying disc degeneration may pose a risk factor for CES [3,9], researchers have emphasized the importance of studying the potential association between CSM and CES [9,36]. Mild adverse events related to CSM, such as transient soreness, are accepted to be common and occur in 23-83% of patients [37]. However, less is known regarding the potential for CSM to cause CES, as the existing literature on the topic is mostly derived from individual case reports [3,4,9].

• Please note that we are not suggesting physical therapy evaluation may increase or decrease the risk of CES. This is noted in the Discussion.

Comment#3

The authors should state more explanations for performing your study.

• Response: Thank you, we agree. Please see the added paragraphs above. We point out that such a large study has not been conducted yet on the topic, with most of the literature being case reports. Our objectives paragraph at the end of the introduction should now have more context as we provide more thorough background on the topic.

Other changes 

1. We added the number of patients (54,846) when describing a study we cite in the introduction. This helps give context, as the study had less patients than ours, and that our study builds upon the past work with having a greater sample size.

2. We deleted a statement from the “Setting and data source” section and combined it with the earlier description of TriNetX in the Methods as it was redundant with our improved ethics statement.

3. We deleted two phrases regarding “claims” data, one in the abstract and another in the Methods in the “Setting and data source” section, as we felt this term was potentially misleading. TriNetX is predominantly based on health records / medical records data. While TriNetX does include claims data as well, our study did not query claims data as we were not focusing on events such as cost.

References (for response document)

1. Topaloglu U, Palchuk MB. Using a federated network of real-world data to optimize clinical trials operations. JCO Clin Cancer Inform. 2018;2:1–10. 

2. Boucher P, Robidoux S. Lumbar disc herniation and cauda equina syndrome following spinal manipulative therapy: A review of six court decisions in Canada. J Forensic Leg Med. 2014;22:159–69. 

3. Haldeman S, Rubinstein SM. Cauda Equina Syndrome in Patients Undergoing Manipulation of the Lumbar Spine. Spine. 1992;17:1469–73. 

4. Tamburrelli FC, Genitiempo M, Logroscino CA. Cauda equina syndrome and spine manipulation: case report and review of the literature. Eur Spine J. 2011;20:128–31. 

5. Hebert JJ, Stomski NJ, French SD, Rubinstein SM. Serious adverse events and spinal manipulative therapy of the low back region: a systematic review of cases. J Manipulative Physiol Ther. 2015;38:677–91. 

6. Terrett A. Misuse of the literature by medical authors in discussing spinal manipulative therapy injury. J Manipulative Physiol Ther. 1995;18:203–10. 

7. Wenban AB. Inappropriate use of the title’chiropractor’and term’chiropractic manipulation’in the peer-reviewed biomedical literature. Chiropr Osteopat. 2006;14:16. 

8. Yang S-D, Chen Q, Ding W-Y. Cauda Equina Syndrome Due to Vigorous Back Massage with Spinal Manipulation in a Patient with Pre-Existing Lumbar Disc Herniation: A Case Report and Literature Review. Am J Phys Med Rehabil. 2018;97:e23–6. 

9. Oliphant D. Safety of spinal manipulation in the treatment of lumbar disk herniations: a systematic review and risk assessment. J Manip Physiol Ther 2004 Mar-Apr273197-210. 2004; 

10. Oppenheim JS, Spitzer DE, Segal DH. Nonvascular complications following spinal manipulation. Spine J Off J North Am Spine Soc. 2005;5:660–6; discussion 666-667. 

11. Nissen T, Wynn R. The clinical case report: a review of its merits and limitations. BMC Res Notes. 2014;7:1–7. 

12. Hutchins TA, Peckham M, Shah LM, Parsons MS, Agarwal V, Boulter DJ, et al. ACR Appropriateness Criteria® Low Back Pain: 2021 Update. J Am Coll Radiol. 2021;18:S361–79. 

13. Pokora RM, Le Cornet L, Daumke P, Mildenberger P, Zeeb H, Blettner M. Validation of Semantic Analyses of Unstructured Medical Data for Research Purposes. Gesundheitswesen Bundesverb Arzte Offentlichen Gesundheitsdienstes Ger. 2020;82:S158–64. 

14. Legnar M, Daumke P, Hesser J, Porubsky S, Popovic Z, Bindzus JN, et al. Natural Language Processing in Diagnostic Texts from Nephropathology. Diagnostics. 2022;12:1726. 

15. Venkatesan M, Uzoigwe CE, Perianayagam G, Braybrooke JR, Newey ML. Is cauda equina syndrome linked with obesity? J Bone Joint Surg Br. 2012;94-B:1551–6. 

16. Cushnie D, Urquhart JC, Gurr KR, Siddiqi F, Bailey CS. Obesity and spinal epidural lipomatosis in cauda equina syndrome. Spine J. 2018;18:407–13. 

17. Ahad A, Elsayed M, Tohid H. The accuracy of clinical symptoms in detecting cauda equina syndrome in patients undergoing acute MRI of the spine. Neuroradiol J. 2015;28:438–42. 

18. Schoenfeld AJ, Bader JO. Cauda equina syndrome: An analysis of incidence rates and risk factors among a closed North American military population. Clin Neurol Neurosurg. 2012;114:947–50. 

19. Vanood A, Singh B, Wheeler K, Patino G. Conus Medullaris Syndrome and Cauda Equina Syndrome after Lumbar Epidural Steroid Injection: A Systematic Review of Literature (4325). Neurology. 2021;96. 

20. Kaiser R, Krajcová A, Waldauf P, Srikandarajah N, Makel M, Beneš V. Are There Any Risk Factors Associated with the Presence of Cauda Equina Syndrome in Symptomatic Lumbar Disk Herniation? World Neurosurg. 2020;141:e600–5. 

21. Shapiro S. Medical realities of cauda equina syndrome secondary to lumbar disc herniation. Spine. 2000;25:348–52. 

22. Korse NS, Pijpers JA, van Zwet E, Elzevier HW, Vleggeert-Lankamp CLA. Cauda Equina Syndrome: presentation, outcome, and predictors with focus on micturition, defecation, and sexual dysfunction. Eur Spine J. 2017;26:894–904. 

23. Fuso FAF, Dias ALN, Letaif OB, Cristante AF, Marcon RM, Barros Filho TEP de. Epidemiological study of cauda equina syndrome. Acta Ortop Bras. 2013;21:159–62. 

24. Ouzts NE, Himelfarb I, Shotts BL, Gow AR. Current state and future directions of the National Board of Chiropractic Examiners. J Chiropr Educ. 2020;34:31–4. 

25. Chang M. The Chiropractic Scope of Practice in the United States: A Cross-Sectional Survey. J Manipulative Physiol Ther. 2014;37:363–76. 

26. Bednarz EM, Lisi AJ. A survey of interprofessional education in chiropractic continuing education in the United States. J Chiropr Educ. 2014;28:152–6. 

27. Ciolek DE, Hwang W, others. Short-Term Alternatives for Therapy Services (STATS) task order: CY 2008 outpatient therapy utilization report. Baltimore: Centers for Medicare & Medicaid Services. Retrieved from http …; 2010. 

28. Himelfarb I, Hyland J, Ouzts N, Russell M, Sterling T, Johnson C, et al. National Board of Chiropractic Examiners: Practice Analysis of Chiropractic 2020 [Internet]. Greeley, CO: NBCE; 2020 [cited 2020 Nov 7]. Available from: https://www.nbce.org/practice-analysis-of-chiropractic-2020/

29. Salsbury SA, Goertz CM, Twist EJ, Lisi AJ. Integration of Doctors of Chiropractic Into Private Sector Health Care Facilities in the United States: A Descriptive Survey. J Manipulative Physiol Ther. 2018;41:149–55. 

30. Adams J, Peng W, Cramer H, Sundberg T, Moore C, Amorin-Woods L, et al. The Prevalence, Patterns, and Predictors of Chiropractic Use Among US Adults: Results From the 2012 National Health Interview Survey. Spine. 2017;42:1810–6. 

31. Harwood KJ, Pines JM, Andrilla CHA, Frogner BK. Where to start? A two stage residual inclusion approach to estimating influence of the initial provider on health care utilization and costs for low back pain in the US. BMC Health Serv Res. 2022;22:694. 

32. Beliveau PJH, Wong JJ, Sutton DA, Simon NB, Bussières AE, Mior SA, et al. The chiropractic profession: a scoping review of utilization rates, reasons for seeking care, patient profiles, and care provided. Chiropr Man Ther. 2017;25:35. 

33. Trager RJ, Daniels CJ, Perez JA, Casselberry RM, Dusek JA. Association between chiropractic spinal manipulation and lumbar discectomy in adults with lumbar disc herniation and radiculopathy: retrospective cohort study using United States’ data. BMJ Open. 2022;12:e068262. 

34. Lavy C, Marks P, Dangas K, Todd N. Cauda equina syndrome—a practical guide to definition and classification. Int Orthop. 2022;46:165–9. 

35. Brinjikji W, Diehn F, Jarvik J, Carr C, Kallmes DF, Murad MH, et al. MRI findings of disc degeneration are more prevalent in adults with low back pain than in asymptomatic controls: a systematic review and meta-analysis. Am J Neuroradiol. 2015;36:2394–9. 

36. Funabashi M, French SD, Kranenburg HA (Rik), Hebert JJ. Serious adverse events following lumbar spine mobilization or manipulation and potential associated factors: a systematic review protocol. JBI Evid Synth. 2021;19:1489–96. 

37. Swait G, Finch R. What are the risks of manual treatment of the spine? A scoping review for clinicians. Chiropr Man Ther. 2017;25:37.

---

## [Decision Letter · Decision Letter 1]

6 Feb 2024

Association between chiropractic spinal manipulation and cauda equina syndrome in adults with low back pain: Retrospective cohort study of US academic health centers

PONE-D-23-39784R1

Dear Dr. Trager,

We’re pleased to inform you that your manuscript has been judged scientifically suitable for publication and will be formally accepted for publication once it meets all outstanding technical requirements.

Kind regards,

Shabnam ShahAli, Ph.D.

Academic Editor

PLOS ONE

Additional Editor Comments (optional):

Reviewers' comments:

Reviewer's Responses to Questions

**Comments to the Author**

1. If the authors have adequately addressed your comments raised in a previous round of review and you feel that this manuscript is now acceptable for publication, you may indicate that here to bypass the “Comments to the Author” section, enter your conflict of interest statement in the “Confidential to Editor” section, and submit your "Accept" recommendation.

Reviewer #1: All comments have been addressed

Reviewer #4: All comments have been addressed

Reviewer #5: All comments have been addressed

2. Is the manuscript technically sound, and do the data support the conclusions?

Reviewer #1: Yes

Reviewer #4: Yes

Reviewer #5: Yes

3. Has the statistical analysis been performed appropriately and rigorously? 

Reviewer #1: Yes

Reviewer #4: Yes

Reviewer #5: Yes

4. Have the authors made all data underlying the findings in their manuscript fully available?

Reviewer #1: Yes

Reviewer #4: Yes

Reviewer #5: Yes

5. Is the manuscript presented in an intelligible fashion and written in standard English?

Reviewer #1: Yes

Reviewer #4: Yes

Reviewer #5: Yes

6. Review Comments to the Author

Reviewer #1: The discussion part was reasonably and appropriately rewrite according to the assumptions of the authors and all other requested corrections are made precisely.

Reviewer #4: I have no further comments. The manuscript is eligible for publication following the editors' review.

Reviewer #5: (No Response)

7. PLOS authors have the option to publish the peer review history of their article (what does this mean?). If published, this will include your full peer review and any attached files.

Reviewer #1: No

Reviewer #4: No

Reviewer #5: No

---

## [Editor Report · Acceptance letter]

1 Mar 2024

PONE-D-23-39784R1 

PLOS ONE

Dear Dr. Trager, 

I'm pleased to inform you that your manuscript has been deemed suitable for publication in PLOS ONE. Congratulations! Your manuscript is now being handed over to our production team.

Kind regards, 

on behalf of

Dr. Shabnam ShahAli 

Academic Editor

PLOS ONE